# Immunogenicity of HIV-1 *Env* mRNA and *Env-Gag* VLP mRNA Vaccines in Mice

**DOI:** 10.3390/vaccines13010084

**Published:** 2025-01-17

**Authors:** Qi Ma, Jing Yang, Xiaoguang Zhang, Hongxia Li, Yanzhe Hao, Xia Feng

**Affiliations:** National Key Laboratory of Intelligent Tracking and Forecasting for Infectious Diseases, National Institute for Viral Disease Control and Prevention, Chinese Center for Disease Control and Prevention, Beijing 100052, China; 13179818951@163.com (Q.M.); yangjing@ivdc.chinacdc.cn (J.Y.); zhangxg@ivdc.chinacdc.cn (X.Z.); lihx@ivdc.chinacdc.cn (H.L.)

**Keywords:** HIV-1 vaccine, mRNA vaccine, VLP vaccine

## Abstract

Background: The development of a protective vaccine is critical for conclusively ending the human immunodeficiency virus (HIV) epidemic. Methods: We constructed nucleotide-modified mRNA vaccines expressing HIV-1 Env and Gag proteins. Env–gag virus-like particles (VLPs) were generated through co-transfection with env and gag mRNA vaccines. BALB/c mice were immunized with env mRNA, env–gag VLP mRNA, env plasmid DNA vaccine, or lipid nanoparticle (LNP) controls. HIV Env-specific binding and neutralizing antibodies in mouse sera were assessed via enzyme-linked immunosorbent assay (ELISA) and pseudovirus-based neutralization assays, respectively. Env-specific cellular immune responses in mouse splenocytes were evaluated using an Enzyme-linked immunosorbent assay (ELISpot) and in vivo cytotoxic T cell-killing assays. Results: The Env-specific humoral and cellular immune responses elicited by HIV-1 env mRNA and env–gag VLP mRNA vaccine were stronger than those induced by the DNA vaccine. Specific immune responses induced by the env mRNA vaccine were significantly stronger in the high-dose group than in the low-dose group. Immunization with co-formulated env and gag mRNAs elicited superior cellular immune responses compared to env mRNA alone. Conclusions: These findings suggest that the env–gag VLP mRNA platform holds significant promise for HIV-1 vaccine development.

## 1. Introduction

HIV-1, which causes AIDS, was identified in the early 1980s and has turned into a worldwide epidemic [1]. Antiretroviral therapy (ART) has greatly decreased AIDS-related illness and death worldwide, increasing the life expectancy of those with HIV [2]. However, as access to ART expands, HIV drug resistance inevitably emerges, affecting the therapeutic efficacy [3]. Developing a safe and effective preventive vaccine remains the most promising strategy to control the global HIV epidemic. Over 30 years of dedicated research and numerous clinical trials have not yet resulted in a licensed HIV vaccine. The virus’s high mutation and recombination rates, extensive genetic diversity, limited understanding of immune protection correlates, and the lack of suitable animal models are major contributing factors [4]. Most conventional methods of immunogen delivery do not result in robust or prolonged immunity against HIV. mRNA vaccines have demonstrated their potential for rapid development and effective induction of immune responses. Amid the COVID-19 pandemic (SARS-CoV-2), there are promising new opportunities for novel mRNA-based vaccines to combat a wide array of challenging diseases [5,6,7]. The mRNA vaccine comprises a single-stranded RNA that encodes antigenic proteins, which can be used as a template for antigen production via the cellular protein synthesis process.mRNA vaccines, such as Moderna and BNT/Pfizer developed for COVID-19, stimulate both cellular and humoral immune responses effectively. These vaccines have demonstrated superior protection rates compared to traditional vaccines [8,9,10]. Moreover, mRNA vaccines have the advantage of an exceptionally rapid transition from design to production, taking as little as 66 days [11], in stark contrast to the conventional 10–15-year timeline for vaccine development. The fast development and manufacturing of mRNA vaccines allow immediate responses to emerging viral threats and their mutations [12,13].

The HIV genome produces several structural proteins, with Env being the only viral antigen found on the surface of the HIV-1 virion. Env is a key target for both neutralizing antibodies and cellular immune responses [8,9,10]. The Gag protein, which is a crucial component of the HIV core, encompasses numerous immunodominant epitopes that are highly conserved. Previous research has suggested a correlation between cellular immune responses targeting Gag and reduced levels of viral replication in individuals with untreated HIV-1 infection as well as long-term non-progressors infected with HIV-2. Thus, the Gag antigen is often included in HIV candidate vaccines. Additionally, previous studies have shown that, therefore, the inclusion of the Gag antigen is a common practice in HIV candidate vaccines. Moreover, previous research has demonstrated that when lentiviral Gag proteins are expressed, they can induce the release of virus-like particles (VLPs) containing Gag with an average size ranging from 100 to 120 nm [14]. These VLPs could display viral Env glycoproteins on their surface when both Env and Gag are co-expressed within the same cell. Virus-like particles (VLPs) are authentic viral particles that self-assemble and lack infectivity but possess antigens on their surfaces. This characteristic enhances immune responses against viral infections by stimulating both humoral and cell-mediated immunity [15,16,17,18]. Their size (<200 nm) facilitates uptake by dendritic cells (DCs), triggering robust cellular immune responses [19]. The antigens present on the repetitive structures of VLPs enhance cross-linking with B cell receptors [20]. This process drives somatic hypermutation of B cells and class-switch recombination from IgM to IgG in immunoglobulins [21]. VLPs promote the differentiation of B cells into plasma cells [22], induce B cell activation through TLR signaling, and enhance the overall production of IgG antibodies [23]. The VLPs are recognized and taken up by DCs, which triggers the maturation process of DCs. This results in the production of pro-inflammatory factors such as TNF-α and IL-1β [24]. Gag-VLPs can serve as carriers for HIV-1 epitopes, glycoproteins, and even Env trimers, enhancing the immunogenicity of antigens. Additionally, previous preclinical investigations have demonstrated that the formation and release of Gag VLPs remain unaffected when combined with HIV-1 epitopes like monomeric gp120 or trimeric gp140 spikes [25,26]. Additionally, clinically approved mRNA vaccines contain N^1^-methylpseudouridine, which reduces the innate immunogenicity of in vitro-transcribed (IVT) mRNA and enhances the specific immune response.

In this study, we constructed nucleoside-modified mRNA vaccines encoding HIV-1 Env and Gag proteins. Env-gag VLPs were generated by co-expressing the Env and Gag proteins when the two mRNA vaccines were administered. The immunogenicity of these vaccines was assessed in BALB/c mice, demonstrating the potential of this platform for HIV-1 vaccine development.

## 2. Materials and Methods

### 2.1. Experimental Materials, Viruses, and Cells

The restriction endonucleases BspQI, 2×Phanta Max MasterMix, T7 High Yield RNA Transcription Kit (N^1^-Me-Pseudo UTP), VAHTSTM DNA Clean Beads, VAHTS RNA Clean Beads, mRNA Cap-2′-O-Methyltransferase, and Vaccinia Capping Enzyme (10 U/μL) were purchased from Nanjing Vazyme Biotechnology Co., Ltd. (Nanjing, China). Restriction endonucleases XhoI and EcoRI were obtained from New England Biolabs (Ipswich, MA, USA). A mouse lymphocyte separation medium was acquired from Dakewe Biotech Co., Ltd. (Beijing, China). Mouse IFN-γ ELISPOT kits were sourced from Mabtech, Nacka Strand, Sweden. Phorbol 12-myristate 13-acetate (PMA) and ionomycin were purchased from SIGMA (St. Louis, MO, USA). The vector plasmid pKMV-α-globin 3′ UTR-polyA was synthesized by Beijing Tianyihuiyuan Biotechnology Co., Ltd. (Beijing, China). HIV-1 AE pseudovirus and HEK293T cells were preserved in the laboratory of Academician Zeng Yi at the National Institute for Viral Disease Control and Prevention, Chinese Center for Disease Control and Prevention. The 2G12 (HIV-1 gp120 specific recombinant human monoclonal antibody from CHO) and Gp120 protein (HIV-1/Clade AE) (Consensus) were purchased from Cambridge Biologics (Brookline, MA, USA). The p24 monoclonal antibody was kindly provided by the NIH AIDS Research and Reference Reagent Program, Germantown, MD, USA.

### 2.2. Construction of Env/Gag Recombinant Plasmid and mRNA

The T7 promoter gene and 5′ UTR of β-globin were introduced into codon-modified HIV-1 env and gag genes via PCR. The env and gag genes used in this study were cloned and modified in our laboratory. The env gene gp145 is a truncated HIV-1 CRF01_AE consensus env gene lacking the intracellular region, while the gag gene is a full-length HIV-1 Subtype B consensus gag; the consensus HIV-1 env and gag genes were acquired and modified according to mammalian codon usage in our laboratory. The codon-optimized genes were synthesized by Invitrogen Corporation Shanghai Representative Office and Shanghai Sangon Biological Engineering Technology & Service Co., Ltd. (Shanghai, China). separately [27,28]. The purified PCR products of the target genes and the vector plasmid pKMV-α-globin 3′UTR-PolyA were digested with Xba I and Xho I, respectively. The digested target gene and plasmid fragments were ligated using T4 DNA Ligase (New England Biolabs, Ipswich, MA, USA) and transformed into competent TOP10 cells. The correct clones were identified by PCR and amplified in Luria-Bertani (LB) medium containing kanamycin (50 µg/mL). Plasmid DNA was extracted and digested with Xho I for identification and sequencing. The identified recombinant plasmids were named plasmids 1 and 2, as shown in Figure 1A. Plasmids were prepared using an endotoxin-free kit purchased from Nanjing Vazyme Biotechnology Co., Ltd., (Nanjing, China), and then linearized. The single digestion system included: 2 μL of restriction enzyme BspQ I, 4 μg of different 5′ UTR recombinant Plasmid 1/Plasmid 2, 2 μL of 10× BspQ I Buffer, and water were added up to 20 μL. Reaction conditions were set at 50 °C for 1 h. The digested products were purified using DNA magnetic beads, and the DNA concentration was measured using a NanoDrop (Thermo Fisher Scientific, Waltham, MA, USA). The linearized products were transcribed in vitro using the T7 High-Yield RNA Transcription Kit (N^1^-Me-Pseudo UTP) purchased from Nanjing Vazyme Biotechnology Co., Ltd. (Nanjing, China). The mRNA products were purified using VAHTS RNA magnetic beads purchased from Nanjing Vazyme Biotechnology Co., Ltd. (Nanjing, China), and the mRNA concentration was measured using a microspectrophotometer. According to the standard kit procedure, the mRNA was capped using the Vaccinia Capping System. The capped products mRNA 1 and mRNA 2 (Figure 1) were purified using VAHTS RNA magnetic beads and quantified using a microspectrophotometer. The final mRNA products were stored at −80 °C for future use.

### 2.3. LNPs Encapsulation of mRNA

The LNP mRNA encapsulation was performed by Shenzhen Yuanxing Gene-tech Co., Ltd. (Shenzhen, China). Briefly, lipid nanoparticles (LNPs) are composed of ionizable lipids, neutral lipids, cholesterol, and PEG-modified lipids, which were dissolved in ethanol and mixed in specific proportions. The mRNA was rapidly mixed with the lipid solution in a microfluidic mixing system to form LNP-mRNA complexes under low pH buffer conditions, which were then stabilized by dilution with a neutral pH buffer. The encapsulated mixture was purified by ultrafiltration to remove the unencapsulated mRNA and free lipids. Dynamic light scattering (DLS) was used to characterize the particle size and dimensions, and the RiboGreen RNA assay was employed to assess the encapsulation efficiency. LNPs encapsulating mRNA1 and mRNA2 were named LNP-env and LNP-gag, respectively.

### 2.4. Expression Verification of Env/Gag mRNA

To verify the expression of in vitro-transcribed mRNA, we seeded HEK293T cells at a density of 2 × 10^5^ cells per well in a 6-well plate. When the cell density reached 80% confluence, the cells were transfected with mRNA1 and mRNA2 using the transfection reagent jetMESSENGER purchased from Polyplus (Illkirch, France). The capped mRNA transcribed from the pKMV-α-globin 3′UTR-PolyA plasmid was used as a negative control. Transfection was performed according to the manufacturer’s instructions. Cells were collected 48 h post-transfection, and the expression of Gp145 and Gag proteins was detected using SDS-PAGE and western blotting.

### 2.5. Indirect Immunofluorescence Assay (IFA)

HEK 293T cells were transfected with mRNA1 and mRNA 2 encoding HIV-1 gp145 and gag, respectively. After 48 h of transfection, the cells were fixed with 4% paraformaldehyde and blocked with 10% fetal bovine serum (FBS) in phosphate-buffered saline (PBS) for 1 h. The cells were subsequently incubated with 2G12 or anti-P24 antibodies, followed by incubation with RBITC-labeled goat anti-human IgG or FITC-labeled goat anti-mouse IgG for 45 min. Cell nuclei were stained with DAPI. Images were captured using confocal microscopy.

### 2.6. Verification of VLP Production by Co-Expression of Env and Gag

1 × 10^6^ cells per well HEK293T cells were grown in six-well plates. When the cell density reached 80%, the cells were co-transfected with mRNA1 and mRNA 2 encoding HIV-1 gp145 and gag using the transfection reagent jetMESSENGER according to the manufacturer’s instructions. At 48 h post-transfection, the supernatant was collected. The supernatant was first centrifuged at 1000 rpm and filtered through a 0.45 µM PES filter. The filtered supernatant was then subjected to ultracentrifugation at 65,000× *g* at 4 °C for 2 h to capture the VLPs. After centrifugation, the supernatant was discarded, and the VLP pellets were resuspended in PBS and stored at 4 °C for further analysis using negative stain electron microscopy (NSEM) or P24 enzyme-linked immunosorbent assay (ELISA).

### 2.7. Mouse Immunization and Detection Protocols

Twenty-five female BALB/c mice, aged 4 to 6 weeks, were acquired from SPF (Beijing, China) Biotechnology Co., Ltd. and housed in the animal facility at the National Institute for Occupational Health and Poison Control, Chinese Center for Disease Control and Prevention. The mice were randomly divided into five groups, with five mice per group. The groups included low-dose LNP-env, high-dose LNP-env, env-gag VLP, pVR-gp145 plasmid DNA vaccine control, and LNP control groups. Immunization was performed at weeks 0 and 3. The vaccine doses for each immunization were 2.5 μg LNP-env/100 μL/head (low-dose LNP-env), 10 μg LNP-env/100 μL/head (high-dose LNP-env), 2.5 μg LNP-env + 2 μg LNP-gag (molar ratio 1:1)/100 μL/head, 100 μg pVR-gp145/100 μL/head, and LNP control/100 μL/head. These were administered intramuscularly at the tibialis anterior muscles. At week 4, mouse sera were collected and stored at −20 °C for antibody detection. Splenic lymphocytes were isolated using a mouse lymphocyte separation solution to detect cellular immune responses. This study was reviewed and approved by the Ethics Committee for Experimental Animals of the National Institute for Viral Disease Control and Prevention, Chinese Center for Disease Control and Prevention (approval number: bdbs20240723052).

### 2.8. Enzyme-Linked Immunosorbent Assay (ELISA)

HIV-1 Gp120 binding antibodies in immunized mouse sera were measured by indirect ELISA. Briefly, 96-well microtiter plates were coated with 100 ng/well purified HIV-1/Clade AE Gp120 protein and incubated overnight at 4 °C. The wells were blocked with PBS containing 5% skim milk for 2 h at 37 °C. Serially diluted sera samples were added and incubated for 1 h at 37 °C. After washing the plates five times, 100 μL of 1:20,000 diluted HRP-conjugated goat anti-mouse IgG antibodies (Zhongshan Golden Bridge, Beijing, China) was then added, and the plates were incubated at 37 °C for 1 h. After that, samples were developed with TMB (WanTai, Beijing, China) at 37 °C for 30 min. The reactions were then stopped using 1 M H_2_SO_4_, and the results were analyzed at a wavelength of 450 nm with a reference wavelength of 630 nm (BioRad, Hercules, CA, USA). Endpoint antibody titers were determined to have the highest serum dilution, where the absorption at 450 nm was more than twice the background signal.

### 2.9. Pseudovirus Based Neutralization Assay

Mouse sera were heat-inactivated at 56 °C for 30 min, and 20 TCID_50_ of HIV-1 AE pseudovirus was mixed with 2-fold serially diluted sera (initial dilution 1:10) in the presence of 40 μg/mL DEAE-dextran. The mixture was incubated for 1 h at 37 °C. The virus-serum mixture was then added to 6 × 10^3^ TZM-bl cells and incubated at 37 °C for 48 h. The cells were lysed, and luciferase activity was measured. The assay was performed in duplicates. The virus input included cells infected with the virus inoculated in the medium instead of the test sera. Luciferase activity in the virus input wells without sera was defined as 100%, and the neutralization activity of the test sera was calculated as the reduction in luciferase activity in the presence of sera.

### 2.10. Enzyme Linked Immunospot (ELISPOT) Assay

The frequency of Env/Gag-specific IFN-γ-secreting cells was assessed using the IFN-γ ELISPOT assay kit (Mabtech AB, Nacka, Sweden) following the manufacturer’s guidelines. In summary, splenic lymphocytes from mice were isolated using a mouse lymphocyte separation medium (Dakewei, Beijing, China). Freshly isolated splenic lymphocytes, at a concentration of 2 × 10^5^ cells, were stimulated with 2 μg/mL of H-2d-restricted Env/Gag-specific cytotoxic T lymphocyte (CTL) epitope peptides in triplicate wells. These cells were then cultured for 48 h at 37 °C and 5% CO_2_ in a 96-well plate pre-coated with purified anti-mouse IFN-γ monoclonal antibodies. Cells cultured in 1% dimethyl sulfoxide (DMSO, Sigma, St. Louis, MO, USA) were used to access the background, whereas cells stimulated with 25 ng/mL of phorbol myristate acetate (PMA, Sigma, St. Louis, MO, USA) and 1 μg/mL of ionomycin (Sigma, St. Louis, MO, USA) were used as positive controls. Following incubation, spots were generated in accordance with the manufacturer’s guidelines. The plates were then air-dried, and the spots were quantified using an Immunospot Reader (CTL, Cleveland, OH, USA). Peptide-specific IFN-γ ELISPOT responses were deemed positive if they exceeded the negative control by at least four times and if the spot-forming cells (SFCs) were greater than 50 SFCs per 10^6^ splenic lymphocytes.

### 2.11. In Vivo Cytotoxicity Assay

Target cells were isolated from the spleens of naive BALB/c mice and incubated at 37 °C for 2 h, either with 2 μg/mL of Env + Gag dominant peptides or in a medium alone. Following the washing step, the peptide-pulsed cells were stained with 5 μM carboxyfluorescein diacetate succinimidyl ester (CFSE^high^), whereas the cells that were not pulsed with peptides were stained with 0.5 μM CFSE (CFSE^low^) in phosphate-buffered saline (PBS) at 25 °C for 15 min. Following washing, 10^7^ CFSE^high^ cells and 10^7^ CFSE^low^ cells were combined in phosphate-buffered saline (PBS) and administered intravenously to either naïve or test mice. The recipient mice were humanely euthanized 12 h after cell transfer, and the transferred splenocytes were extracted from the spleens of these mice for flow cytometry analysis. The percentage of specific killing was determined using the following method: the ratio of non-peptide-treated control spleen cells to peptide-sensitized spleen cells is defined as the percentage of CFSE^low^ cells divided by the percentage of CFSE^high^ cells. The percentage-specific lysis (% killing) is subsequently calculated as 100 × [1 − (the ratio of cells recovered from naive mice/the ratio of cells recovered from test mice)].

### 2.12. Data Analysis

The humoral immune level was represented as the geometric mean titer of HIV Gp120 binding antibodies or neutralizing antibodies in mouse sera. The specific cellular immune responses measured by ELISPOT are expressed as the average number of HIV Env/Gag-specific IFN-γ spot-forming cells (SFCs) per million spleen lymphocytes (SFCs/10^6^ lymphocytes). Specific cellular immune responses measured by in vivo cytotoxic T cell-killing assays were expressed as a percentage of specific lysis. Data are presented as absolute mean value +/− the standard error of the mean (SEM). Statistical analysis was performed using GraphPad Prism software (version 7.0; GraphPad Software, La Jolla, CA, USA), and statistical differences between the experimental and control groups were analyzed using two-way analysis of variance (ANOVA) and considered statistically significant at *p* < 0.05.

## 3. Results

### 3.1. Construction and Characterization of HIV mRNA Vaccine

As illustrated in Figure 1A, two HIV-1 mRNA vaccines encoding the Env and Gag proteins were successfully constructed. Just as Electron microscopy images of LNP-env and LNP-gag vaccines Figure 1B, the LNPs encapsulating the env and gag mRNA had diameters of 110nm and 108 nm, respectively (Figure 1C). The polydispersity index (PDI), a measure of the width of the molecular weight distribution, was below 0.1 for both vaccines (Env: 0.06, Gag: 0.04, Figure 1C), indicating a homogeneous distribution of nanoparticles. Encapsulation efficiencies were greater than 80% (96% for Env and 84.2% for Gag; Figure 1C). The expression of Env and Gag proteins in both mRNA vaccines was confirmed via western blot analysis. As shown in Figure 1E, bands of 145 kDa and 55 kDa were detected in HEK293T cells transfected with env and gag mRNA. IFA results further demonstrated that the Env protein (Gp145) was expressed on the cell membrane, while the Gag protein was expressed intracellularly (Figure 1F). These findings suggest that both in vitro-transcribed mRNAs can efficiently express the inserted genes in eukaryotic cells. Overall, the results shown in Figure 1 validate the successful encapsulation, uniform particle size, and structural stability of the LNP-mRNA vaccines.

### 3.2. Formation and Identification of Virus-like Particles

Co-expression of Env and Gag within cells facilitated the production of VLPs (Figure 2A). To verify VLP formation, gp145 and gag mRNA at different molar ratios were co-transfected into HEK293T cells. The VLPs were concentrated from the supernatant by ultracentrifugation, and the P24 content was detected by ELISA (Figure 2B). The morphology of the concentrated VLPs post-ultracentrifugation was analyzed using transmission electron microscopy (TEM). The VLPs appeared roughly spherical, with an average diameter of approximately 90 nm (Figure 2C). There was no significant difference in P24 content among VLPs produced by the co-transfection of gp145 and gag mRNA at molar ratios of 4:1, 2:1, and 1:1. Therefore, for mouse immunization, a 1:1 molar ratio of gp145 and gag LNP-mRNA was used in the VLP group.

### 3.3. Strong Env-Specific Humoral Immune Responses Elicited by HIV-1 Env mRNA and Env–Gag VLP mRNA Vaccines in BALB/c Mice

BALB/c mice were immunized twice at 3-week intervals with 2.5 μg LNP-env, 10 μg LNP-env, 2.5 μg LNP-env +2.0 μg LNP-gag, 100 μg DNA vaccine, or 100 μL LNP control. (Figure 3A). At week 4, sera were collected, and gp120-specific binding antibodies were measured using indirect ELISA (Figure 3B). Mice immunized with the DNA vaccine displayed low levels of Gp120 binding antibodies, and the geometric mean anti-Gp120 IgG titer was 1:332.3 ± 16.1. In contrast, LNP-env and env–gag VLP mRNA vaccines elicited significantly higher titers: 1:82,269 ± 2676 (2.5 μg LNP-env), 1:157,519 ± 6058 (10 μg LNP-env), and 1:84,082 ± 6467 (2.5 μg LNP-env + 2.0 μg LNP-gag). These titers were 250–500 times higher than those induced by the DNA vaccine. A dose-dependent increase in binding antibody titers was observed for the mRNA vaccines. The neutralizing activity of the mouse serum was tested using a pseudovirus-based neutralization assay. The initial dilution of the tested mouse sera was 1:10, and no neutralizing activity was found at this dilution in the DNA vaccine and LNP control groups. HIV-1 AE specific neutralizing antibodies were induced in env mRNA and env–gag VLP mRNA vaccines, with 50% neutralizing antibody titers as follows: 1:23.9 ± 2.1 (2.5 μg LNP-env), 1:45.7 ± 3.47 (10 μg LNP-env), and 1:26.6 ± 2.4 (2.5 μg LNP-env + 2.0 μg LNP-gag). Neutralizing antibody titer correlated positively with the vaccine dose for the mRNA vaccine (Figure 3C).

### 3.4. Potent Env/Gag-Specific Cellular Immune Responses Detected by IFN-γ ELISPOT Assay

Similar to antibody detection, the mice splenic lymphocytes were isolated at week 4, and the frequency of Env/*Gag*-specific *IFN-*γ secretion cells was detected by IFN-γ ELISPOT assay. As shown in Figure 3D, low levels of Env-specific *IFN-*γ secretion cells were detected in mice immunized with DNA vaccine (285 ± 72 SFCs/million splenocytes), while env mRNA and env–gag VLP mRNA Vaccines elicited extremely high frequency of Env-specific *IFN-*γ secretion cells. The frequency was 3647 ± 83 SFCs/million splenocytes (2.5 μg LNP-env), 4623 ± 181 SFCs/million splenocytes (10 μg LNP-env), and 4159 ± 68 SFCs/million splenocytes (2.5 μg LNP-env + 2.0 μg LNP-gag). Env-specific *IFN-*γ secretion cells were positively correlated with vaccine dose for mRNA vaccine. Additionally, more *IFN-*γ secretion cells induced by env–gag VLP mRNA vaccines were comparable with env mRNA only when the same dose of LNP-env was used, which was similar to that of the 10 μg LNP-env mRNA vaccine group. Furthermore, for the env–gag VLP mRNA vaccine group, the frequency of (Env+Gag)-specific IFN-γ secretion cells reached 4589 ± 150 SFCs/million splenocytes.

### 3.5. Potent Env/Gag-Specific Cellular Immune Responses Detected by In Vivo Cytotoxicity Assay

To evaluate whether the Env/Gag-specific cellular immunity induced by the aforementioned vaccines not only exhibited significant ex vivo activity but also demonstrated distinct cytolytic functions in vivo. For this purpose, an in vivo cytotoxicity assay was conducted, where equal quantities of Env/Gag peptide-pulsed and unpulsed target splenocytes were intravenously injected into vaccinated mice prior to detection. Flow cytometry analysis conducted after 12 h showed that all vaccine groups effectively targeted and killed the Env/Gag peptide-pulsed population. As shown in Figure 4A,B, consistent with the ELISPOT results, the percentage of specific lysis in the DNA vaccine group was the lowest at only 19%. In mice immunized with the env mRNA and env–gag VLP mRNA vaccines, the percentage of specific lysis was over 90% (94.1% in the 2.5 μg LNP-env group and 97.3% in the 10 μg LNP-env group), though the difference between these two groups was not statistically significant. The highest level of specific lysis was observed in the env–gag VLP mRNA vaccinated group (99.1%).

## 4. Discussion

In previous studies, we demonstrated the feasibility of incorporating the 5′UTR of β-globin in mRNA vaccines [29]. The current study explores the immunogenicity of HIV mRNA and VLP vaccines in mice, building on this foundation. We constructed mRNA vaccines encoding HIV-1 Env and Gag proteins and immunized mice with either env mRNA or env–gag VLP mRNA. Our results show that both vaccines effectively induced strong HIV-specific humoral and cellular immune responses.

Developing a global HIV/AIDS vaccine involves the necessity of eliciting Env-specific humoral immune responses to block viral entry. Env is the sole viral protein present on the surfaces of both viral particles and cells that are infected. As such, it is a critical target for neutralizing antibodies that block viral entry and antibody-dependent cell-mediated cytotoxicity (ADCC), which eliminates virus-infected cells. However, the HIV envelope protein, Env, is structurally complex and weakly immunogenic, making it challenging to induce the immune system to produce broadly neutralizing antibodies against Env antigens. Due to the presence of the gp41 intracellular tail, the low expression of the Env protein is also considered one of the mechanisms of immune evasion by HIV/SIV viruses. In a previous study, we removed the intracellular domain of the HIV Env protein and obtained the gp145 gene, which includes full-length gp120 and the extracellular region of gp41 (partially transmembrane). This modification retains all extracellular antigenic epitopes and reduces cytotoxicity. Compared to soluble antigens, membrane-anchored Env proteins can be expressed on the host cell surface, facilitating the recognition and binding of antigens by B-cell surface receptors, thereby stimulating B-cell activation, proliferation, and differentiation. In a study by Melzi E, membrane-bound immunogens encoded by mRNA significantly increased the activation and recruitment of neutralizing antibody precursors to germinal centers and lowered their affinity threshold [30]. Our experiments demonstrated that these constructs induced much higher titers of both binding and neutralizing antibodies compared to DNA vaccines. During the natural process of viral infection, HIV-1 Env proteins are expressed on the surface of HIV-1 viral particles in the immune system. By co-expressing env and gag mRNA, we generated VLPs that mimic the natural HIV infection process, thereby enabling the dual presentation of Env proteins on the cell surface and VLPs. This strategy enhances B cell activation and stimulates higher antibody titers. VLPs enhance the presentation of antigens in a conformation that mimics pathogens, thereby promoting recognition by B cells. This approach not only masks poorly neutralizing epitopes but also enhances access to highly neutralizing epitopes, ultimately leading to the induction of broadly neutralizing antibodies(bnAbs) [31]. However, in our study, there was no statistically significant difference in antibody titers between the env mRNA and env+gag VLP groups, even at equivalent doses of env mRNA. This finding is consistent with those of Zhang et al. [32]. The low incorporation rate of Env into HIV-1 viral particles during the simulation of natural infection might be a factor, as high-density protein display is crucial for effective B cell receptor (BCR) cross-linking. BCR cross-linking aids B-cell proliferation, antibody affinity maturation, and the production of bnAbs [33]. This phenomenon might also explain why the vesicular stomatitis virus G protein can induce an effective antibody response with its multiple copies (high density) displayed on viral particles [34]. Compared to DNA vaccines, mRNA vaccines can induce higher antibody titers. On the other hand, the goal of HIV vaccines is to induce cellular immune responses, including Gag-specific CD8+ T cells, to control viral replication and thereby block HIV infection. Conventional vaccines typically consist of specific antigens formulated with adjuvants to induce adaptive immune responses. Adjuvants are essential for amplifying innate immune responses, which, in turn, facilitate the activation of T cells [35,36]. Our study revealed that a low dose of the VLP mRNA vaccine induced strong cellular immune responses comparable to those induced by a high dose of the env-mRNA vaccine. Unlike mRNA or DNA vaccines, VLPs can directly activate the MHC I-CD8 T cell pathway without the requirement for extracellular antigens. Similarly, mannose-glycosylated Rabbit Hemorrhagic Disease Virus (RHDV) VLPs have been shown to function as MHC I-class foreign antigens, effectively stimulating cytotoxic T cells [37]. Cytotoxic T lymphocytes (CTLs) are key effector cells involved in specific immune responses, capable of recognizing and killing host cells infected with viruses by releasing cytotoxic proteins such as perforin and granzymes, leading to target cell apoptosis. This is crucial for controlling HIV infections. Our in vivo CTL assay using the CFSE labeling method showed that after primary and booster immunizations, the specific killing rate in the VLP group exceeded 99%, demonstrating its potential as a promising HIV vaccine candidate.

Our study had some limitations. First, while our vaccines induced robust immune responses, they did not generate high titers of broad-spectrum neutralizing antibodies. One potential method for improving neutralizing antibody induction is sequential germline targeting, which guides the immune system toward the development of bnAbs. However, this approach may require multiple immunizations or complex regimens. In future studies, we aim to evaluate the effects of multi-vector sequential and repeated immunization using mRNA and other vector platforms. This research will provide valuable experimental data to inform subsequent preclinical and clinical trials.

## 5. Conclusions

To summarize, we successfully developed an mRNA vaccine encoding HIV-1 Env and Gag proteins, which induced robust humoral and cellular immune responses in mice. Compared to DNA vaccines, the mRNA vaccine significantly enhanced the levels of both binding and neutralizing antibodies. While there was no significant difference in antibody titers between the mRNA and virus-like particle (VLP) groups, the vaccine effectively stimulated strong Env/Gag-specific cellular immunity. Future research will investigate sequential immunization strategies and multi-vector approaches to further enhance the production of broadly neutralizing antibodies.

## Figures and Tables

**Figure 1 vaccines-13-00084-f001:**
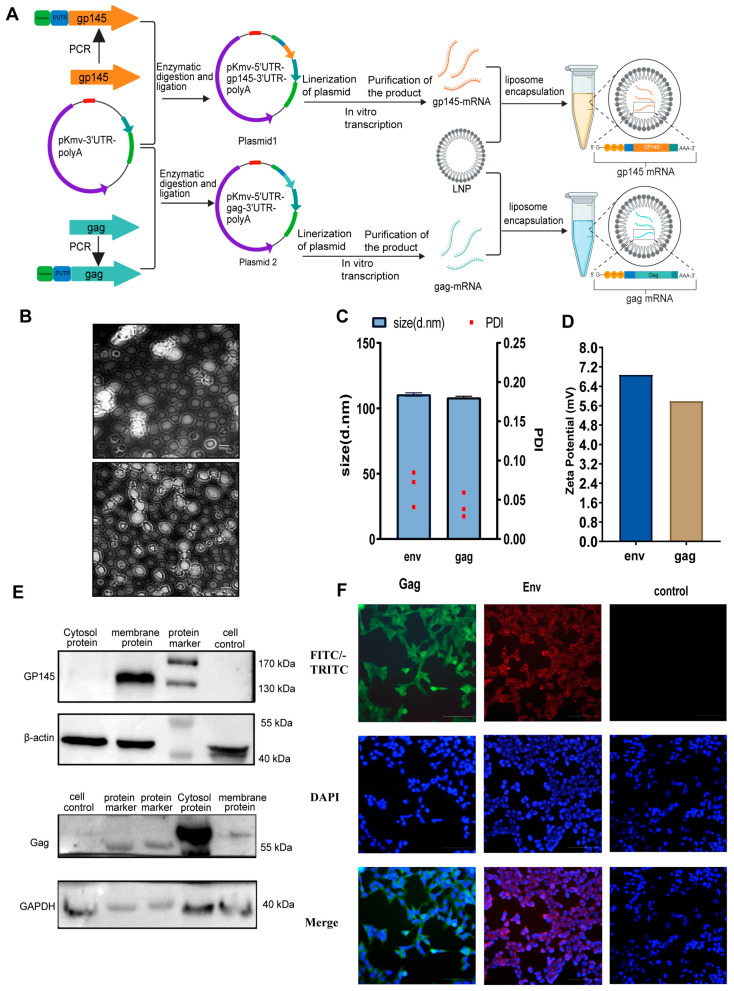
Design and characterization of HIV-1 mRNA Vaccines: (**A**) Schematic representation of the construction of env and gag mRNA vaccines. (**B**) Electron microscopy images of LNP-env and LNP-gag vaccines. Scale bar: 100 nm. (**C**) Results of particle size (blue) and polydispersity index (red dots) analysis for LNP-env and LNP-gag vaccines. (**D**) Zeta potential measurement results for LNP-env and LNP-gag vaccines. (**E**) Detection of protein expression of env and gag mRNA in the cell membrane and cytoplasm of HEK293T cells. (**F**) Immunofluorescence analysis of Env and Gag protein expression, with ENV and Gag mRNA transfected into HEK293T cells for 48 h. Detection was performed using 2G12 and anti-P24 antibodies as primary antibodies and goat anti-human TRITC and goat anti-human FITC as secondary antibodies. Scale bar: 100 μm.

**Figure 2 vaccines-13-00084-f002:**
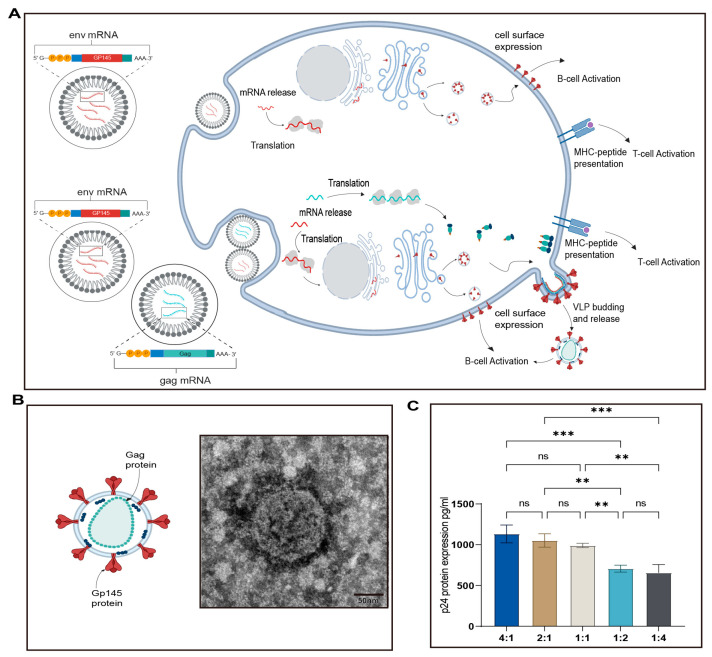
Production and characteristics of Env-Gag VLPs by co-transfection with gp145 and gag mRNA: (**A**) A comparison of the mechanisms of action between mRNA vaccines (**top**) and virus-like particle (VLP) vaccines (**bottom**) within cells. (**B**) Schematic representation of VLPs (**left**) and electron microscopy images of VLPs (**right**). The VLPs were concentrated using ultracentrifugation and prepared for negative staining in electron microscopy. (**C**) ELISA results show the expression of Gag protein in VLPs produced by co-transfecting cells with gag and env mRNA at different molar ratios. Scale bar: 50 nm. ** *p* < 0.01; *** *p* < 0.001; ns, no significant difference.

**Figure 3 vaccines-13-00084-f003:**
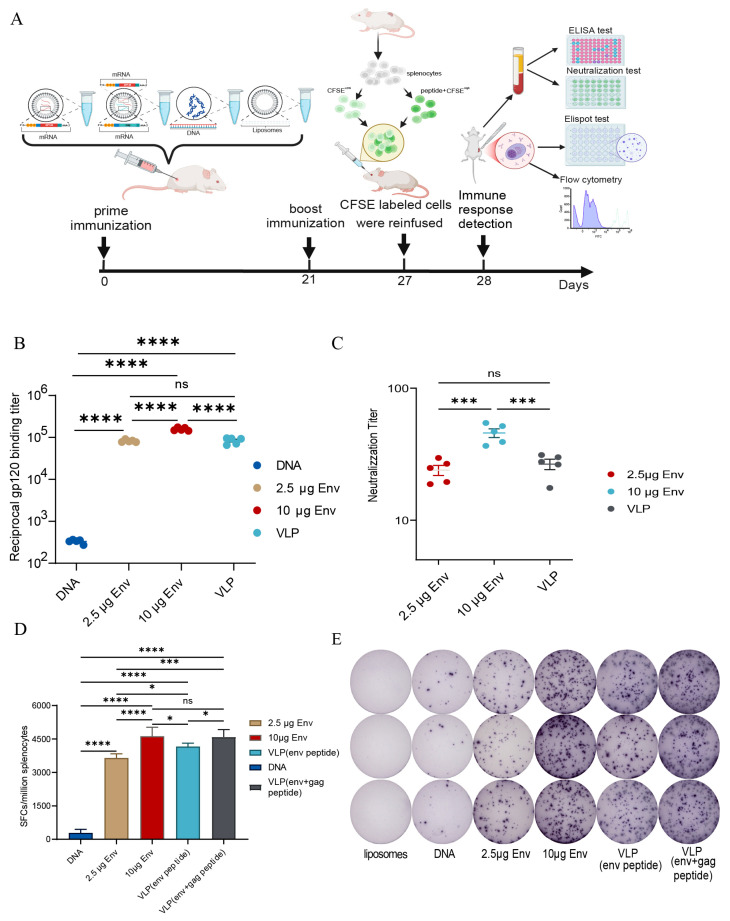
Immunogenicity of HIV mRNA Vaccine and Virus-Like Particle Vaccine in Mice (**A**) Immunization and detection timeline on day 0 and day 21. Mice (*n* = 5) were immunized with LNP-env, env-gag VLP, DNA vaccine control, or LNP control. On day 27, lymphocytes isolated from the spleens of naive BALB/c mice were stimulated with Env+Gag specific peptides and used as specific target cells. Splenocytes from the same source without specific stimulation served as internal controls. These two populations were labeled with different concentrations of CFSE and were injected intravenously into naïve or test mice. On day 28, mice sera and splenocytes were isolated for analysis. (**B**) Indirect ELISA was used to detect the titers of HIV Gp120-specific IgG in mice sera. (**C**) The 50% neutralizing antibody titers against the HIV-1 pseudovirus strain AE14 in mice sera. (**D**,**E**) IFN-γ ELISPOT assay was employed to assess T-cell immune responses in mice splenocytes. 5 × 10^4^ freshly isolated splenic lymphocytes were stimulated in duplicate wells with 2 μg/mL of Env/Gag-specific peptides, and the IFN-γ secretion was detected. The mean numbers of IFN-γ spot formation cells (SFCs) per million splenic lymphocytes of each group are shown. Five mice in each group were analyzed. Error bars represent SEM for each group * *p* < 0.05; *** *p* < 0.001; **** *p* < 0.0001; ns, no significant difference.

**Figure 4 vaccines-13-00084-f004:**
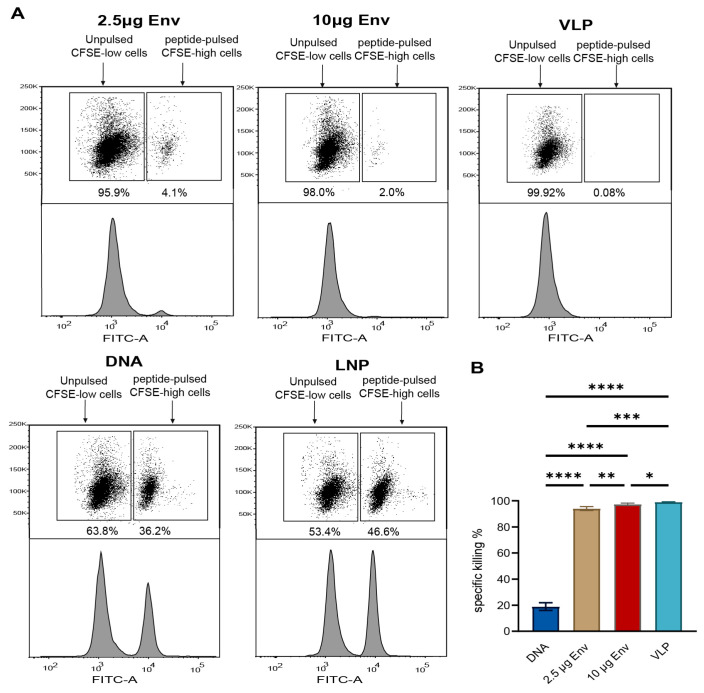
In vivo cytotoxicity assay: One week post the last immunization, the killing of i.v.-transferred Env/Gag-pulsed splenocytes was monitored by flow cytometry. (**A**) Representative dot plots and histograms of CFSE^low^ (unpulsed) and CFSE^high^ (Env/Gag-pulsed) populations, 12 h after transfer (1:1) into vaccinated mice. (**B**) Percentage-specific killing of Env/Gag-pulsed target cells in the vaccinated mice compared to the naive control. CFSE stands for carboxy-fluorescein succinimidyl ester. * *p* < 0.05; ** *p* < 0.01; *** *p* < 0.001; **** *p* < 0.0001.

## Data Availability

All data related to this study are included in this article.

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
