# Peer review of "Immunogenicity of HIV-1 Env mRNA and Env-Gag VLP mRNA Vaccines in Mice"

_vaccines, 2025, doi:10.3390/vaccines13010084_

Round 1
Reviewer 1 Report
Comments and Suggestions for Authors
Authors (Qi Ma et al) developed mRNA and VLP Vaccines against HIV-1 and evaluated the immunogenicity of these vaccines in mice. The title and abstract of the manuscript reflect the research paper, the manuscript is well written; clear, and mostly easy to understand. The work is well discussed and well illustrated. The authors validated successful encapsulation, uniform particle size, and structural stability of the obtained particles. The results of the research scientifically sound. Nevertheless, I have some comments on the manuscript, if the authors make changes to the text, the text would be significantly improved.
1. line 110 kanamycin concentration is not specified, please specify
2. line 103 The source of HIV-1 env and gag genes is not specified, it is not clear whether these were viral DNA or synthesized sequences, specify the source or the manufacturer
3. line 140 Specify the manufacturer
4. line 149. 2G12 and P24 (the source is not described in the M&M section).
5. line 170. It is not clear what low-dose and high-dose mean. Please provide details
6. lines 184-185 it is not clear where gp120 came from, provide the source
There are typos in the text, for example, lines 102, 127, 352, 356, 452, etc.
Author Response
Reviewer 1
Thank you for your detailed and insightful review of our manuscript. We greatly appreciate your positive remarks regarding the relevance and quality of our study. Below, we address your specific comments and suggestions.
1. Line 110 Correction
Reviewer Comment: line 110 kanamycin concentration is not specified, please specify”
Response: Thank you for pointing out this omission. We have carefully addressed your comment and specified the concentration of kanamycin used in the experiment.
The following modification has been made: In Line 110, we have added the kanamycin concentration:“The final concentration of kanamycin used in the experiment was 50 µg/mL.”
2. Line 103 Correction
Reviewer Comment: line 103 The source of HIV-1 env and gag genes is not specified, it is not clear whether these were viral DNA or synthesized sequences, specify the source or the manufacturer.
Response: Thank you for highlighting this important point. We acknowledge the lack of clarity regarding the source of the HIV-1 env and gag genes and have addressed this issue in the revised manuscript.
The following modification has been made: In Line 103, we have clarified the source of the genes:
HIV-1 subtype AE env genes were cloned from the infected paid blood donors in Beijing, and the consensus sequence based on these prevalent strains was obtained by aligning. The codons of the consensus env sequence were modified according to mammalian codon usage.
HIV-1 subtype B gag genes were cloned from the infected paid blood donors in Henan, and the consensus sequence based on these prevalent strains was obtained by aligning. The codons of the consensus gag sequence were modified according to mammalian codon usage.
3. Reviewer Comment: line 140 Specify the manufacturer
Response: Thank you for your observation. We have updated the manuscript to include the manufacturer information as requested.
The following modification has been made: In Line 140, we have specified the manufacturer as follows: When the cell density reached 80% confluence, the cells were transfected with mRNA1 and mRNA 2 using the transfection reagent jetMESSENGER(Polyplus)
4. Reviewer Comment: line 149. 2G12 and P24 (the source is not described in the M&M section). lines 184-185 it is not clear where gp120 came from, provide the source
Response: Thank you for pointing out the omission regarding the source of 2G12 gp120 and P24. We have now clarified this in the Materials and Methods section.
The following modification has been made: In the Materials and Methods section, we have specified the source of these reagents as follows: The 2G12 monoclonal antibody and The gp120 protein were purchased from Cambridge Biologics. The p24 monoclonal antibody was produced by the HIV-1 p24 Hybridoma (183-H12-5C) cell line, which was obtained from Dr. Bruce Chesebro.in line 100-102
5. Reviewer Comment: . line 170. It is not clear what low-dose and high-dose mean. Please provide details
Response: Thank you for highlighting this point. We understand the need to clearly define the terms "low-dose" and "high-dose" for clarity and reproducibility.
The following modification has been made: In the revised manuscript, we have specified the definitions of "low-dose" and "high-dose" as follows: “Low-dose refers to a concentration of 2.5 μg LNP-env /100μL/head, while high-dose refers to a concentration 10μg LNP-env /100μL/head."
6. Reviewer Comment: There are typos in the text, for example, lines 102, 127, 352, 356, 452, etc.
Response: Thank you for pointing out the typos in the text. We have carefully reviewed the manuscript and corrected all identified typos, including those on lines 102, 127, 352, 356, 452, and others. Additionally, we conducted a thorough proofreading of the entire manuscript to ensure accuracy and consistency throughout the text.
Below are examples of the corrections made:
Line 102: [changed "condon" to "codon"]
Line 127: [changed " ," to ", "]
Line 352: [changed " IFN-γsecretion " to " IFN-γ secretion "]
Line 356: [changed " comaprable " to " comparable "]
Line 452: [A period (.) was added after "antigens.]
Reviewer 2 Report
Comments and Suggestions for Authors
Good manuscript.
introduction is correct according to the objectives and lets to understand critícal points of the virus and of the vaccines.
Methods are well explaind but some suggestions about the organization of the groups and sample size are maded (see maniscript).
Results and discussion are also clear explained. Figures in results are very good.

Author Response
Thank you for your thorough and insightful review of our manuscript. We appreciate your positive remarks regarding the relevance and quality of our study. Below, we address your specific comments and suggestions:
1.Reviewer Comment: It should be interesting to explain these numbers,why 25?, criteria to organize groups with 5 animals?
Response: Thank you for your thoughtful question. We appreciate the opportunity to clarify our rationale for these experimental design choices.
1.Why 25?
The total number of 25 animals was chosen to ensure sufficient statistical power for the analysis of group differences while adhering to ethical principles of minimizing the use of animals in research. This number allowed us to distribute animals across multiple experimental groups while maintaining adequate sample sizes for meaningful statistical comparisons.
2.Criteria to organize groups with 5 animals:
Groups of 5 animals were selected based on standard practices in preclinical studies, balancing statistical requirements with ethical considerations. A group size of 5 provides a sufficient sample to detect significant differences in response variables, as determined by power analysis conducted prior to the study. Additionally, the group size aligns with recommendations for minimizing animal use while obtaining robust and reproducible results.
2.Reviewer Comment: Is it enough to take samples only one moment along time to evalúate the vaccine?
Response: Thank you for raising this important question. Our decision to take samples at a single time point was based on findings from our previous study[1], which demonstrated that the immune response reached its peak at 1 week post last vaccination. This informed our rationale for selecting this specific time point to focus on evaluating the vaccine’s efficacy at its most relevant stage.
We acknowledge that additional sampling at multiple time points would provide a more comprehensive understanding of the vaccine’s dynamics, including the kinetics of immune response development and duration. This is an important aspect we plan to address in future studies.
3.Reviewer Comment: What about the response along the time? Results came from a single simple 4 weeks after vaccination. I suggest to comment it.
Response: Thank you for your insightful comment. We agree that evaluating the immune response over time would provide a more comprehensive understanding of the vaccine's dynamics. In this study, we chose to focus on the 4-week time point because our previous research demonstrated that immune responses peaked at this stage post-vaccination. This decision allowed us to assess the vaccine’s efficacy at its most relevant and robust point. We acknowledge that monitoring the immune response at multiple time points would yield valuable information on the kinetics and durability of the immune response. To address this, we plan to conduct long-term monitoring in future experiments to evaluate the vaccine’s immunogenicity over an extended timeframe.
Conclusion
We thank both reviewers for their valuable comments and suggestions, which have helped us improve our manuscript. We have incorporated all recommended changes to enhance the robustness and impact of our study. We look forward to your further feedback.
REFERENCES
1. Xiaozhou, H.; Jing, Y.; Hongxia, L.; Yanzhe, H.; Xia, F. Study on rAd5F35-SIVenvT vaccine in combination with rMVA-SIVenvT vaccine in mice. Chinese Journal of Microbiology and Immunology 2021 -06 -30